# miR-154-5p Is a Novel Endogenous Ligand for TLR7 Inducing Microglial Activation and Neuronal Injury

**DOI:** 10.3390/cells13050407

**Published:** 2024-02-26

**Authors:** Hugo McGurran, Victor Kumbol, Christina Krüger, Thomas Wallach, Seija Lehnardt

**Affiliations:** 1Charité—Universitätsmedizin Berlin, Einstein Center for Neurosciences Berlin, 10117 Berlin, Germany; hugo.mcgurran@charite.de (H.M.); victor.kumbol@charite.de (V.K.); 2Institute of Cell Biology and Neurobiology, Charité—Universitätsmedizin Berlin, Corporate Member of Freie Universität Berlin, Humboldt-Universität zu Berlin and Berlin Institute of Health, 10117 Berlin, Germany; christina.krueger@charite.de (C.K.); thomas.wallach@charite.de (T.W.); 3Department of Neurology, Charité—Universitätsmedizin Berlin, Corporate Member of Freie Universität Berlin, Humboldt-Universität zu Berlin and Berlin Institute of Health, 10117 Berlin, Germany

**Keywords:** microRNA, neurodegeneration, neuroinflammation, toll-like receptor, neurons, microglia

## Abstract

Toll-like receptors (TLRs) are a collection of pattern recognition sensors that form a first line of defence by detecting pathogen- or damage-associated molecular patterns and initiating an inflammatory response. TLR activation in microglia, the major immune cells in the brain, can trigger the release of inflammatory molecules, which may contribute to various CNS diseases including Alzheimer’s disease. Recently, some microRNAs were shown to serve as signalling molecules for TLRs. Here, we present miR-154-5p as a novel TLR7 ligand. Exposing microglia to miR-154-5p results in cytokine release and alters expression of the TLR signalling pathway dependent on TLR7. Additionally, miR-154-5p causes neuronal injury in enriched cortical neuron cultures and additive toxicity in the presence of microglia. Finally, intrathecal injection of miR-154-5p into mice leads to neuronal injury and accumulation of microglia in the cerebral cortex dependent on TLR7 expression. In conclusion, this study establishes miR-154-5p as a direct activator of TLR7 that can cause neuroinflammation and neuronal injury, which may contribute to CNS disease.

## 1. Introduction

In recent years, the role of the innate immune system in both initiating and propagating neurodegenerative and neuroinflammatory diseases, such as Alzheimer’s disease (AD), has been widely acknowledged, but the mechanistic breadth of innate immunity in neuronal damage is still unclear. One key part of the innate immune system are the toll-like receptors (TLRs). These are a group of highly conserved pattern recognition receptors that detect pathogen-associated molecules such as lipopolysaccharide (LPS), double-stranded DNA and RNA, or molecules released from host-derived cells, such as heat shock proteins, among others.

TLRs are expressed by all innate immune cells including microglia, the main immune cells in the brain, but are also expressed in various non-immune cells, such as endothelial cells and neurons [1,2]. Due to their ubiquity, TLRs form a vital part of innate immunity and help facilitate the crosstalk between the innate and adaptive immune system [3], which can facilitate progression from acute to chronic inflammation. Therefore, their activation contributes to many diseases, including neurological disease [3,4,5]. The TLR family encompasses 10 members in humans and 13 in mice, localising to either the cell surface (TLR1, 2, 4, 5, 6, 10) or to endosomes (TLR3, 7, 8, 9). TLR7 is responsible for the detection of single-stranded RNA (ssRNA) and is therefore typically involved in the defence of viral infections. However, as TLR7 also has the capability to detect endogenous ssRNA such as microRNAs (miRNAs) [6], it is plausible that TLR7 activation via miRNAs may influence inflammation in neurodegenerative disease. We have previously shown that TLR7 contributes to neurodegenerative and neuroinflammatory processes in the context of AD [6]. Also, recent data have demonstrated that TLR7 activation is involved in the neurodegeneration induced by certain viruses [7,8]. Additionally, inhibition of TLR7 can reduce acute inflammation from surgically induced cognitive deficits [9]. However, deletion of *Tlr7* can rescue some cognitive deficits in AD mice but does not affect overall inflammation or amyloid-β deposition [10]. Together, these data indicate that TLR7 has a function in neurological disease, particularly in neurodegeneration, but the specific outcome of receptor activation may be highly dependent on the animal model and disease context.

MiRNAs are non-coding ssRNAs that are ~22 nucleotides long whose primary role is to regulate post-transcriptional gene expression [11]. MiRNAs are critical controllers of gene and protein expression, and any dysregulation will end up affecting many biological processes [11]. So, unsurprisingly, miRNAs have been associated with a wide range of diseases, such as cancer [12], cardiovascular disease [13], autoimmune disease [14], and neurological disease [15]. The canonical role of miRNAs in suppressing messenger RNA is a likely mechanism of action in these diseases. However, we showed in previous work that the miRNA *let-7b* has a non-canonical function, i.e., does not exert its primary effects through gene regulation at the post-transcriptional level, acting as a ligand for TLR7, and has implications as a signalling molecule in the CNS for neuroinflammation and neurodegeneration [6].

The molecular mechanisms of the binding between miRNAs and TLR7 are still under investigation. However, studies on TLR7 crystal structures revealed that uridine-rich ssRNAs can activate a second binding site on TLR7, which primes the guanosine-specific first binding site for synergistic activation [16]. Hence, there is some degree of sequence specificity to TLR7 binding as uridine- and guanosine-rich or other specific motifs within miRNAs preferentially activate TLR7 to induce cytokine release [17,18]. In addition to *let-7b*, a few other miRNAs have been discovered that act as TLR7 ligands, including miR-298-5p and miR-100-5p [19,20]. Furthermore, miRNAs have been shown to regulate microglial inflammation by interacting with TLR signalling. For instance, while miR-155 is required for LPS (TLR4 agonist)-mediated IL6 production in microglia, miR-124 can assist in shifting microglia polarisation from an M1 (pro-inflammatory state induced by LPS or IFN-γ with the expression of pro-inflammatory cytokines) to an M2 (anti-inflammatory state induced by IL-4/IL-13 with resolution of inflammation and tissue repair) state after cerebral ischemia. miR-154-5p was shown to reverse a circ-Scmh1-induced M2 polarisation [21,22,23].

MiR-154-5p has been identified as a possible therapeutic target not only in certain cancers [24,25], but also in neurological diseases such as neuropathic pain and glioblastoma [26,27]. However, the detailed mechanistic role of miR-154-5p in neurological diseases, and particularly neurodegenerative diseases, is still unresolved. Early data indicates that this miRNA is at least partially dysregulated in both AD and Parkinson’s patients [28,29]. The implications of such dysregulation are currently unknown. In vascular dementia, exosomal miR-154-5p is overexpressed, and its inhibition can prevent endothelial progenitor cell dysfunction [30]. However, miR-154-5p also prevents induced M2 microglial polarisation in BV2 cells [23] and reduced neuronal apoptosis in a cell model of epilepsy by reducing TLR5 expression [31], indicating that miR-154-5p’s function may be context dependent. As miR-154-5p is associated with neuroinflammation [23,32], and TLR7 senses miRNAs [6], we sought to investigate the role of miR-154-5p acting as a signalling molecule for this receptor in the CNS in the context of microglial activation and neurodegeneration.

In this study, we identify miR-154-5p as an endogenous ligand for TLR7. We demonstrate that extracellular miR-154-5p activates microglia in vitro and in vivo. Also, this miRNA causes both cell-autonomous and microglia-mediated neuronal toxicity in vitro and in vivo. Together with the previous data indicating the dysregulation of miR-154-5p in AD [28], our data reveal that miR-154-5p is a valid candidate for further study in the neurodegenerative context and will contribute to a better understanding of the role miRNAs may play in neurodegeneration.

## 2. Methods

### 2.1. Mice

C57BL/6 and *Tlr*7^−/−^ mice were bred at the FEM, Charité—Universitätsmedizin Berlin, Germany. *Tlr*7^−/−^ mice were generously provided by S. Akira (Osaka University, Osaka, Japan). Animals were maintained according to the guidelines of the committee for animal care. All animal procedures were approved by the Landesamt für Gesundheit und Soziales (LAGeSo) Berlin, Germany.

### 2.2. Cell Lines

HEK-Blue™ Secreted Embryonic Alkaline Phosphatase (SEAP) cells expressing human/mouse TLR7 or human/mouse TLR8, as well as the respective control cell lines HEK-Blue™ Null1-k, Null2-k, Null1, and Null1V (Invivogen, San Diego, CA, USA), were cultured in Dulbecco’s modified Eagle’s medium (DMEM; Invitrogen #41965062, Carlsbad, CA, USA). DMEM was supplemented with 10% heat-inactivated fetal calf serum (FCS, Gibco #10082–147, Thermo Fisher Scientific, Waltham, MA, USA) and penicillin (100 U/mL)/streptomycin (100 μg/mL), both obtained from Gibco #15140–122, Thermo Fisher Scientific, Waltham, MA, USA. All experiments with cell lines were conducted on cells with fewer than 20 passages. Cells were treated with Blasticidin, Zeocin, and Normocin as per the manufacturer’s instructions to prevent the growth of bacteria, fungi, and mycoplasma.

### 2.3. Primary Cell Culture

*Neurons.* Primary cell cultures of neurons were generated from the cortex of embryonic E17.5 mice. The brain was extracted into HBSS buffer without calcium or magnesium (Gibco #14170088). Subsequently, the meninges, superficial blood vessels, cerebellum, and optical lobe were removed from the cortices. The cortices were then homogenised and incubated with 0.5 mL 2.5% trypsin (Gibco #15090046) for 20 min at 37 °C. The trypsin was quenched with heat-inactivated FCS (Gibco #10082147). The tissue was washed with HBSS buffer with calcium and magnesium (Gibco #24020117) twice. After the second wash, the HBSS buffer was reduced to approximately 2 mL, to which 100 μL DNase (Roche #1284932001) was added for 1 min. Next, the tissue was washed twice with 10 mL neurobasal media (Gibco #21103049) supplemented with 1% L-Glutamine (Gibco #25030024), 1% penicillin/streptomycin (Gibco #15140122), and 2% B27 supplement (Gibco #17504044). The volume was brought up to 10 mL, and the tissue was resuspended and centrifuged for 4 min at 17 rcf. The supernatant was collected, then centrifuged for 5 min at 264 rcf. The pellet was then resuspended in neurobasal media and seeded onto PDL-coated coverslips in a 24-well plate at a density of 5 × 10^5^ and placed in an incubator at 37 °C in humidified air with 5% (*v*/*v*) CO_2_. The following day, a half-media change was performed with neurobasal media. Experiments were started after 3 days in vitro.

*Microglia.* Primary microglia were generated from the cortex of P1-4 mice. The meninges, superficial blood vessels, cerebellum, and optical lobe were removed. The cortices were then homogenised with 3 mL 2.5% trypsin for 25 min at 37 °C. The reaction was stopped with 5 mL FCS. The tissue was then resuspended, and 100 μL DNase was added for 1 min. The tissue was then centrifuged at 264 rcf for 5 min. Supernatant was discarded, and the pellet was resuspended in DMEM (Gibco, #41965062) supplemented with 1% pen/strep (Gibco, #15140122) and 10% FCS. The cells were then passed through a 70 μM cell strainer, added to T75 flasks, and cultured at 37 °C in humidified air with 5% (*v*/*v*) CO_2_. The following day, a full media change was performed, and the cells were cultured for 10–14 days in 12 mL DMEM with an additional 5 mL DMEM added on day 7.

*Co-cultures of neurons and microglia.* For co-culture experiments, on day 3 in vitro, half of the medium of the neuronal cultures (containing 1 × 10^5^ cells) was replaced with DMEM containing 60,000 microglia for an approximate 1:8 ratio of microglia to neurons. The experiments were started the following day.

### 2.4. Tumor Necrosis Factor Enzyme-Linked Immunosorbent Assay (ELISA)

Primary murine microglia (3 × 10^4^ cells in a 96-well plate) were incubated with lyovec-complexed miRNAs for the indicated time and with the indicated concentration. The supernatants were collected and analysed for TNF according to the manufacturer’s instructions (Invitrogen, #88732488).

### 2.5. HEK-Blue TLR7/8 Activation Assay

HEK293 cells expressing human/mouse TLR7/8 NF-κB/Secreted Embryonic Alkaline Phosphatase (SEAP) reporter and their respective parental control cell lines were used for TLR activation assays. All cell lines were purchased from Invivogen. A total of 30,000 cells were seeded onto 96-well plates in DMEM. The following day, the media was changed to 90% HEK-Blue detection reagent (InvivoGen #hb-det2) and 10% DMEM. The cells were then incubated with indicated miRNAs (10 μg/mL) complexed with Lyovec, or controls for 24 h. All conditions were performed in triplicate and analysed at a wavelength of 655 nM using a SpectraMax iD3 (Molecular Devices, San Jose, CA, USA).

### 2.6. Reverse Transcription Quantitative Polymerase Chain Reaction (RT-qPCR)

Primary microglia or neurons were stimulated with the indicated miRNA or control for 24 h. The RNA was then collected using a commercial kit (Qiagen, #74104, Hilden, Germany) according to the manufacturer’s instructions. cDNA was then made using M-MLV reverse transcriptase (Promega, Madison, WI, USA)**.** SYBR-green qPCR analysis was performed using the StepOnePlus RT-qPCR system (Applied Biosystems, Waltham, MA, USA). Data are expressed using 2^–∆∆Ct^. All conditions were performed in triplicate, and the average CT value was used. Primers used are listed below (Table 1).

### 2.7. Synthetic Oligonucleotides and Cell Stimulation

Oligonucleotides were purchased from Integrated DNA Technologies. Oligonucleotides were modified with a 5′ phosphorylation and phosphorothioate bonds in every base. The control oligonucleotide is a scrambled sequence of let-7b with no homology to any sequence in mouse or human [6]. The sequences for the control oligonucleotide and miR-154-5p are UGAGGUAGAAGGAUAUAAGGAU and UAGGUUAUCCGUGUUGCCUUCG, respectively. For experiments testing HEK cells, microglia, and co-cultures of neurons and microglia, synthetic oligonucleotides were complexed with Lyovec (Invivogen, lyec-rna) according to the manufacturer’s protocol. Neuron monocultures were stimulated with uncomplexed oligonucleotides. Positive controls LPS (Enzo Life Sciences 0111:B4, Farmingdale, NY, USA), loxoribine (Invivogen, tlrl-lox), R848 (Invivogen, tlrl-r848), and TNF (PeproTech, 315-01A, Cranbury, NJ, USA) were applied without Lyovec.

### 2.8. Immunochemistry

For immunocytochemistry, cells were washed, then fixed in 4% PFA for 20 min. Cells were then stained with primary antibodies (1:500) in staining buffer (PBS with 2% normal goat serum (NGS), 0.2% Triton-X) for NeuN (Millipore, MAB377, Burlington, MA, USA) or IBA1 (Wako 01919741, Tokyo, Japan) overnight at 4 °C. Cells were washed again and stained with secondary antibodies (Alexa fluor 488 and 568, 1:500) for 1 h at room temperature in the dark. Nuclei were stained with DAPI (1:10,000) for 1 min at room temperature and sealed with immumount (Epredia, Kalamazoo, MI, USA). TUNEL apoptosis assays were performed using In Situ Cell Death Detection kit TMRred or Fluorescein following the manufacturer’s instructions.

For immunohistochemistry, brains were fixed in 4% PFA then cut at 14 μM thickness and stored at −70 °C until use. When ready for staining, sections were treated with 4% PFA for 15 min, washed, then blocked and permeabilised in blocking buffer (PBS with 5% NGS and 0.2% Triton-X) for 1 h. Sections were then stained with NeuN (Millipore, MAB377 or ABN78), IBA1 (Wako 01919741), MAP2 (Millipore AB5622 or MAB3418), or Neurofilament (Millipore, MAB5262) antibody at 1:500 dilution in staining buffer overnight at 4 °C. Sections were washed again and incubated with secondary antibodies (Alexa Fluor 488 and 568, Invitrogen) for 1 h at room temperature in the dark. Sections were then washed and counterstained with DAPI for 1 min at room temperature and finally sealed with immumount.

### 2.9. Microscopy and Imaging

Imaging was performed on an Olympus IX81 microscope. For analysis of NeuN, Iba-1, and TUNEL staining in toxicity assays, 6 images were taken for each coverslip at 40× magnification, and images were then analysed in FIJI (v. 1.54f) and data normalised to the unstimulated control variable. For tissue sections, 3 images on the left and right hemisphere at 20× for Iba-1, and 40× or 60× for NeuN positivity at approximately an interaural distance of 1.86 mm were analysed. For the quantification of cortical NeuN-positive cells in tissue sections, the analysis was performed by two independent examiners, and the data subsequently pooled for analysis.

### 2.10. Intrathecal Injection

Intrathecal injection into mice was performed as described previously [6,33]. Briefly, 10 μg of RNA or water was injected into the intrathecal space. Seventy-two hours later, the mice were transcardially perfused with 4% PFA, and the brains were removed and cryoprotected in a 30% sucrose solution. The brains were cut coronally (14 μM) and thaw-mounted on glass slides.

### 2.11. Statistics

All data are expressed as the mean ± SEM. Statistical differences over all groups were analysed using a one-way ANOVA followed by Dunnett’s test. Statistical analysis of two specific groups was performed using a two-tailed Student’s *t*-test. Intrathecal injection experiments were analysed using a two-way ANOVA followed by Tukey’s test to compare all groups and account for treatment condition and genetic background. Statistics were calculated in Graphpad Prism 9.0 (Graphpad Software, Dotmatics, Boston, MA, USA). 

## 3. Results

### 3.1. miR-154-5p Is an Endogenous Ligand for TLR7 and TLR8

TLR7 and TLR8 are responsible for the detection of viral ssRNA [34]. However, it has been shown that endogenous ssRNAs such as miRNAs are also able to activate TLR7/8 in a sequence-dependent manner [7]. First, we utilised the machine learning software Braindead (version 5.0.7), which uses sequence and structure analysis of TLR7 and the respective candidate miRNA to predict TLR7/8 activation likelihood [35]. We found that miR-154-5p had a high probability of TLR7 activation (braindead score: 90.1%).

To validate this, we used an HEK-Blue Secreted Embryonic Alkaline Phosphatase (SEAP) reporter cell assay overexpressing human or mouse TLR7 or TLR8, as well as their respective parental control lines. The cells were incubated with synthetic miR-154-5p, control oligonucleotide, which contains a mutated *let-7b* sequence (containing six nucleotide exchanges in the central and 3′ regions, outside of the 5′ *let-7* seed sequence that is important for post-transcriptional silencing) and was previously shown to not activate TLR7/8 [6], or the TLR7 and TLR7/8 agonists loxoribine and R848. We observed that miR-154-5p significantly activated mouse TLR7 and human TLR7/8 (Figure 1A–C) but not mouse TLR8 (Figure 1D).

To confirm the functional relevance of this TLR7/8 activation ability, we isolated and exposed wild-type (WT) and *Tlr*7^−/−^ primary murine microglia to miR-154-5p for 24 h and measured the TNF content in the supernatant via ELISA. MiR-154-5p induced significant TNF release in WT but not *Tlr*7^−/−^ microglia (Figure 1E) and did so in a dose-dependent manner (Figure 1F). An amoeboid shape of WT microglia was observed in response to incubation with miR-154-5p, loxoribine, and LPS, indicating an activated state. In contrast, treatment of microglia lacking TLR7 with miR-154-5p did not cause such morphological changes (Figure 1G). Overall, these data show that extracellular miR-154-5p serves as a ligand of mouse and human TLR7 as well as human TLR8.

### 3.2. miR-154-5p Alters the TLR7 Signalling Pathway in Microglia

Microglia are the primary component of the CNS’s immune system and represent a major TLR-utilising cell type. As such, the activation of TLRs in microglia results in an extensive intracellular signalling cascade. To test whether the activation of mouse TLR7 in HEK reporter cells corresponds to alterations in the TLR7 signalling pathway of primary mouse microglia, we tested for changes in the TLR signalling cascade by stimulating WT and *Tlr*7^−/−^ microglia with miR-154-5p for 24 h and subsequently analysing select transcripts by RT-qPCR (Figure 2).

We observed that, in both WT and *Tlr*7^−/−^ microglia, expression levels of the TLR adaptor proteins *TRIF* and *MYD88* were mostly unchanged relative to unstimulated cells, whereas both *SARM1* and *IRAK4* were relatively downregulated in WT but not *Tlr*7^−/−^ microglia. *P65* was slightly decreased compared to the control in WT microglia, possibly indicating a lesser involvement of the NF-κB pathway induced by miR-154-5p. Downstream signalling components such as *IRF7* and the cytokine RNA expression of *IFNβ*, *TNF*, and particularly *IL6* (with almost a 1000-fold increase in expression) were all greatly increased in WT but not *Tlr*7^−/−^ microglia. Taken together, these data demonstrate that miR-154-5p serving as a signalling molecule for TLR7 specifically modulates the TLR-linked signalling cascade in microglia.

### 3.3. Extracellular miR-154-5p Induces Neuronal Injury In Vitro

The activation of multiple TLRs, including TLR2, 4, 5, and 7, has been shown to induce neuronal injury [20,36,37,38]. Activation of these receptors in microglia leads to the release of cytokines and chemokines, which, in high concentrations, are toxic to neurons, [39]. Neurons also express TLR7, although they do not release high levels of cytokines upon activation [40]. Thus, we tested whether miR-154-5p is able to cause neuronal injury and whether this effect is cell-autonomous or mediated by microglia.

In purified primary cortical neuronal cultures (Appendix A), extracellular miR-154-5p caused neuronal injury, as assessed by a significant loss of NeuN-positive cells. This neurotoxic effect was dependent on time and dose (Figure 3A). The relative levels of TUNEL- to NeuN-positive cells, giving an indication of the rate of apoptosis, showed a modest increase in apoptosis at lower concentrations of miR-154-5p, but this was not detected at the highest dose (Figure 3B). To determine whether the loss of neurons and increase in neuronal apoptosis observed in purified cortical neuron cultures (Figure 3C) were mediated by TLR7, we performed a similar experiment testing primary *Tlr*7^−/−^ neurons (Figure 3D). No changes in neuronal numbers at 5 or 8 days post stimulation of miR-154-5p at the highest dose (10 μg/mL) compared to the control were observed.

Although we found that miR-154-5p causes cell-autonomous neurotoxicity, as miR-154-5p also strongly activates microglia, we questioned whether the presence of microglia would induce an additive effect of neuronal toxicity. Therefore, we co-cultured primary neurons and microglia (Appendix A) and exposed the co-cultures to miR-154-5p for 5 days. We observed a dose-dependent reduction in neuronal viability (Figure 3E). This effect was relatively greater than that observed in neuron-enriched cultures without microglia (40% NeuN loss vs. 15% NeuN loss) in the same 5-day period (Figure 3F), indicating an additive neurotoxic effect from the presence of microglia.

### 3.4. Intrathecal Injection of miR-154-5p into Mice Triggers Microglial Accumulation and Neuronal Injury via TLR7

To investigate the role of extracellular miR-154-5p as a TLR7 ligand in the CNS in vivo, we injected this miRNA intrathecally into both WT and *Tlr*7^−/−^ mice. Three days post injection, microglial numbers, as assessed by quantifying IBA1-positive cells, were significantly increased in the cerebral cortex of WT mice compared to the control (Figure 4A,B). Importantly, this effect was not detected in the cerebral cortex of *Tlr*7^−/−^ mice (Figure 4A,B), supporting our in vitro data showing that functional TLR7 signalling is required for miR-154-5p-induced effects in mice. Additionally, the neurotoxic effects observed in vitro were confirmed as we found a significant decrease in NeuN-positive cells in the cerebral cortex of WT mice but not in *Tlr*7^−/−^ mice (Figure 4C,D). Overall, these data demonstrate that extracellularly delivered miR-154-5p induces microglial accumulation and neurodegeneration through TLR7 in vivo.

## 4. Discussion

MiRNAs have long been associated with disease due to their primary function in regulating gene expression at the post-transcriptional level. However, since discovering that miRNAs have roles beyond this and act as ligands for immune receptors such as TLR7, their potential implication in both health and disease has been greatly expanded. Here, we demonstrate that miR-154-5p is able to directly activate TLR7 and subsequently induces microglial cytokine release and neuronal injury. MiR-154 has been investigated in a variety of diseases, from neuropathic pain [32] and colorectal cancer [25] to vascular and Alzheimer’s dementia [28,30]. This accentuates the pathological diversity that miR-154, and other miRNAs, could be involved in and highlights the potential miRNAs have for pharmaceutical intervention. Due to the sheer number of miRNAs, it can be speculated that many new non-mRNA interactions will be discovered for miRNAs in the future. This idea has begun bearing fruit, with data showing, for example, that miR-126 can bind to caspase-3 to prevent its dimerisation and subsequent apoptosis or that miR-1 binding directly to ion channels modulates cardiac action potentials [41,42]. However, due to their capability to sense ssRNA, TLR7 and TLR8 are prime candidates for miRNA interaction. Indeed, we and others have shown that miRNAs such as *let-7b*, miR-298, miR-100-5p, miR-4288, and miR-29a-3p [6,19,35,43], and likely many more miRNAs, bind to and directly activate TLR7. Direct activation of TLR7 in neurons by extracellular *let-7*, miR-298, and miR-100, abundantly expressed in the brain, leads to neuronal caspase-3 activation and subsequent neurodegeneration. Also, neuronal TLR7 activation by *let-7* involves MyD88 and IRAK 4 signalling [6]. However, the detailed neuronal signalling cascade downstream of TLR7 directly activated by the different miRNAs remains unresolved. Activation of TLR7 in microglia by *let-7*, miR-100, and miR-298 involves elements of the canonical TLR pathway, including NF-κB transcription and, ultimately, the release of potentially neurotoxic cytokines/chemokines [6,19]. Thus, although a direct interaction between the respective miRNA and TLR7 has been established, the detailed signalling cascades in the various CNS cell types induced by such an interaction are only partially identified. However, the TLRs’ primary role is to act as a first line of defence against pathogens and to recognise host-derived damage-associated molecules, thereby triggering the innate immune system and inducing inflammation, which has now become one of the areas at the forefront of research in neurodegenerative diseases such as AD and Parkinson’s disease [44,45]. By identifying new activators and modulators of the innate immune system, one could access novel therapeutic channels to treat neurodegenerative and/or neuroinflammatory diseases. This is particularly relevant for miRNAs as it would potentially allow us to leverage an endogenous system for the desired outcome, though, as of today, TLR7 is established as the only major receptor recognising miRNAs. Other research has begun uncovering new non-gene targets, for example, miR-1 binds to cardiac membrane proteins [42] and miR-126-5p binds to caspase-3 to regulate autophagy [41]. This provides exciting opportunities to expand the role of miRNAs in disease. In this context, we define extracellular miR-154-5p as a ligand for TLR7, which can induce microglial activation as well as cell-autonomous and microglia-mediated neuronal injury.

In our current study, we demonstrate that miR-154-5p modulates the TLR signalling pathway in microglia and induces TNF release from these cells in a dose- and TLR7-dependent fashion, indicating that extracellular miR-154-5p may shift microglia from a homeostatic state to a more pro-inflammatory state. It has been shown that miR-154-5p is able to reverse a circ-Scmh1-induced microglial polarisation from M1 to M2 via STAT6 binding [23], which may alter TLR7 signalling [46]. Also, miR-154-5p seems to be involved in allergic inflammation mediated by MCP-1 [47]. Although we determined some cytokines in addition to TNF at the mRNA level produced in response to miR-154-5p, an in-depth characterisation of the cytokine/chemokine pattern released from microglia responding to miR-154-5p would be valuable to fully understand the interaction between miR-154-5p and TLR7 as well as the cellular consequences of such an interaction, especially in human and disease contexts. Also, further research is required to characterise the role of miR-154-5p in inducing a microglial M1 and/or M2 state and to assess whether TLR signalling and further signalling cascades, such as MCP-1, which miR-154-5p is considered to regulate [47], are involved in these processes. Finally, the alteration of some signalling components, but not others such as MYD88 and TRIF, is of interest. As outlined above, little is currently known about the effects of miRNAs acting as ligands on the expression of TLR signalling components and what the consequences of such changes are. We suggest that further research will help to unravel the TLR signalling pathways induced by extracellular miRNAs, such as miR-154-5p.

Of note, in addition to TLR7, human TLR8 is also capable of interacting with miRNAs, including miR-154-5p, as shown in our current study. However, murine TLR8 does not respond to ssRNAs in the same way as TLR7 does [48,49]. Thus, in a human context, miR-154-5p may interact with TLR8 to cause currently unknown effects. Furthermore, even though here we show the role of miR-154-5p as a direct activator of TLR7, it is also worth considering that miR-154-5p will still exhibit its canonical role in gene expression. This may influence our results and is important to understand when considering the use of miRNAs as a pharmaceutical target as altering the typically tightly controlled expression of miRNAs may induce unintended side effects.

We showed that miR-154-5p, when applied to co-cultures of neurons and microglia, causes a significant loss of neurons. Much of this effect is likely due to the activation of microglia by miR-154-5p causing a high degree of cytokine and chemokine release. However, we also demonstrated that miR-154-5p causes a basal level of toxicity to neurons in the absence of microglia, indicating that the activation of TLR7 in neurons, as non-immune cells, can still induce potent effects leading to cell-autonomous injury. The exact role of TLR7 in non-immune cells like neurons is still unclear, though some evidence suggests it is less associated with viral immunity, as it is in the case in glial activation, than it is involved in cellular processes such as dendrite growth, neuroexcitability regulation, and cell death [6,50,51]. Still, the exact mechanisms of TLR7-mediated toxicity in neurons remain to be uncovered, particularly in the context of endogenous molecules such as miRNAs acting as TLR ligands.

To confirm whether extracellular delivery of miR-154-5p could lead to microglial activation and neuronal injury in vivo, we directly administered miR-154-5p into the cerebrospinal fluid in both WT and *Tlr*7^−/−^ mice. In line with our in vitro data, we observed a reduction in neuronal numbers in the cerebral cortex of WT mice. Additionally, accumulation of microglial numbers was observed in WT but not *Tlr*7^−/−^ mice. Thus, miR-154-5p can acutely activate microglia in vivo, which may, over a longer period, sufficiently stimulate microglia to cause further neurotoxicity, which might be speculated to be at least one factor contributing to neurodegenerative processes as they occur in AD, in which miR-154 expression is upregulated [28]. The miRNA’s ability to modulate microglial activation in vivo is critical as inflammation is a typical component of neurodegenerative diseases and can be an instigator and/or propagator of neurodegeneration. Future research using miRNA inhibitors and/or TLR7 antagonists in mouse disease models, including AD mouse models, will further elucidate the role of the interaction between miR-154-5p and TLR7 in the context of neurodegeneration. Also, further studies on the function of miR-154-5p, particularly in animal models already exhibiting high inflammation or long-term effects of miR-154-5p over/underexpression, are critical to understand the function miRNAs play in immune regulation. In addition, studies to specifically assess the miR-154-5p kinetics in neurodegeneration and high-throughput screening for modulators of miR-154-5p expression may be valuable strategies for future research.

As the primary immune cells in the CNS, the role of microglia in neurodegeneration is being intensively studied. Even though the microglia’s principal role is to mount immune responses, their role extends into non-immune functions including brain development, synaptic pruning, neuronal plasticity, and programmed cell death [52,53,54,55]. Naturally, this means that the alteration/dysfunction of microglia will have wide-reaching implications in disease, immunologically rooted or not. For instance, Alzheimer’s and vascular dementia are not pathologically exclusive to the immune system, but inflammation still plays a significant role in the pathology [44,56]. Therefore, a modulation of the immune responses could have beneficial clinical outcomes for these diseases. What is currently unknown, however, is how miR-154-5p, or other miRNAs acting as ligands for immune receptors, are behaving kinetically in either healthy or pathologic states. It has been shown that some miRNAs such as *let-7b* are able to enter neurons and microglia and aggregate in their endosomal compartment [6]. However, it is not known whether miRNAs naturally emanate to the extracellular space of the CNS in their native, non-exosomal state under non-pathological conditions, and if so, whether they are actively engulfed into membranes, form complexes, or induce a wider inflammatory cascade. Therefore, a better understanding of the dynamics of miRNAs in their extracellular form is vital to uncovering their functions and is especially relevant under neurodegenerative conditions whereby cell contents are released due to cell injury and death [19]. Finally, in perceiving the development of miRNA-based strategies for use in the clinic, it will be crucial to understand the potential differences between responses induced by different miRNAs. For instance, unbound or encapsulated miRNAs, or antisense oligonucleotides and small molecule inhibitors, may modulate TLR7 responses in differing ways. Further research is required to determine whether there is a sequence-specific nature of miRNAs that induces the effects seen or whether it is a more general TLR7-mediated effect.

Despite miR-154 being upregulated in AD patients [28], it is unknown whether this is a pathological outcome of the disease, where a reduction in miR-154 may be desirable, or whether it is a compensatory mechanism to prevent further pathological damage, where upregulation may be beneficial. Such future questions building upon novel neurotoxic pathways due to miRNA dysregulation require further investigation in both disease-specific animal models and in the clinic to determine the therapeutic capacity of miRNA modulation. Finally, future studies may focus on therapeutic interventions that could modulate miR-154-5p levels or block its interaction with TLR7/8 in neurodegenerative disease.

## 5. Conclusions

Here, we present miR-154-5p as a novel, direct activator of TLR7. Extracellular application of miR-154-5p both in vitro and in vivo leads to activation of microglia through TLR7 signalling and results in neuronal injury. In addition, miR-154-5p acting as a TLR7 ligand induces neuronal cell death in a cell-autonomous fashion, supporting a role for TLR7 mediating toxicity in non-immune cells. However, given the extremely complex nature of neurodegenerative and neuroinflammatory diseases, alongside the discovery of new interaction between miRNA and TLR7, further interdisciplinary research is required to uncover how the intricacies of miRNAs acting as signalling molecules for TLR7 and potentially further immune receptors may impact clinical outcomes and to develop novel therapeutic options.

## Figures and Tables

**Figure 1 cells-13-00407-f001:**
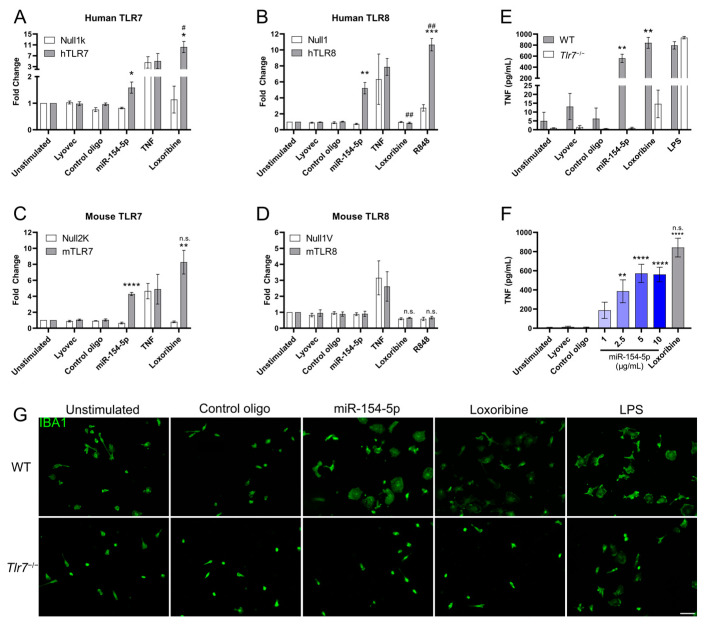
miR-154-5p activates mouse and human TLR7. Human (**A**,**B**) and murine (**C**,**D**) TLR7/8 HEK293-derived reporter and corresponding null control cell lines were stimulated with synthetic miR-154-5p (10 μg/mL) or controls for 24 h. Unstimulated, Lyovec, and control oligo (10 μg/mL) conditions served as negative controls, while TLR7 agonist Loxoribine (1 mM) and TLR7/8 agonist R848 (100 ng/mL) served as positive controls. Results are expressed as the mean ± SEM with *n* = 3. Significance determined using Student’s *t*-test compared to respective control line with same condition (*) or by Student’s *t*-test compared to miR-154-5p (#/n.s). (**E**) TNF levels in the supernatant were assessed by ELISA after 24 h stimulation of both wild-type (WT) and *Tlr7*^−/−^ microglia with Lyovec, control oligo (10 μg/mL), miR-154-5p (10 μg/mL), or Loxoribine (1 mM). Significance determined by Student’s *t*-test relative to *Tlr7*^−/−^ levels for indicated condition. (**F**) Dose response of WT microglia after 24 h stimulation with indicated concentration of miR-154-5p. Significance determined One-way ANOVA followed by Dunnett’s test for multiple comparison compared to unstimulated (*) or by Student’s *t*-test compared to miR-154-5p (n.s). (**G**) Representative images of WT and *Tlr7*^−/−^ microglia immunostained with IBA1 antibody and treated with control oligo (10 μg/mL), miR-154-5p (10 μg/mL), Loxoribine (1 mM), or LPS (100 ng/mL, TLR4 agonist) for 24 h, demonstrating alterations in microglial morphology. Scale bar represents 50 μm. *^/#^
*p* < 0.05, **^/##^
*p* < 0.01, *** *p* < 0.001, **** *p* < 0.0001.

**Figure 2 cells-13-00407-f002:**
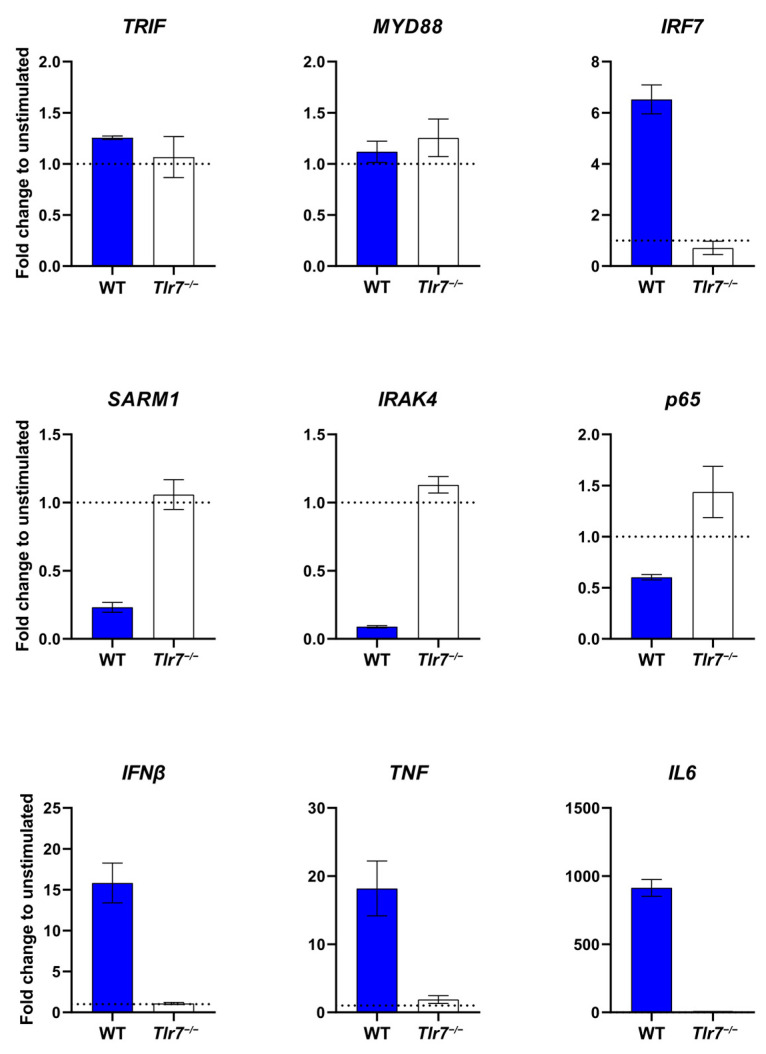
miR-154-5p alters components of the TLR signalling pathway in microglia. WT and *Tlr7*^−/−^ microglia were stimulated with miR-154-5p (10 μg/mL) for 24 h, then analysed for TLR signalling pathway components by RT-qPCR. Data is expressed as the mean fold change ± SEM compared to unstimulated control as indicated by the dashed line at y = 1.

**Figure 3 cells-13-00407-f003:**
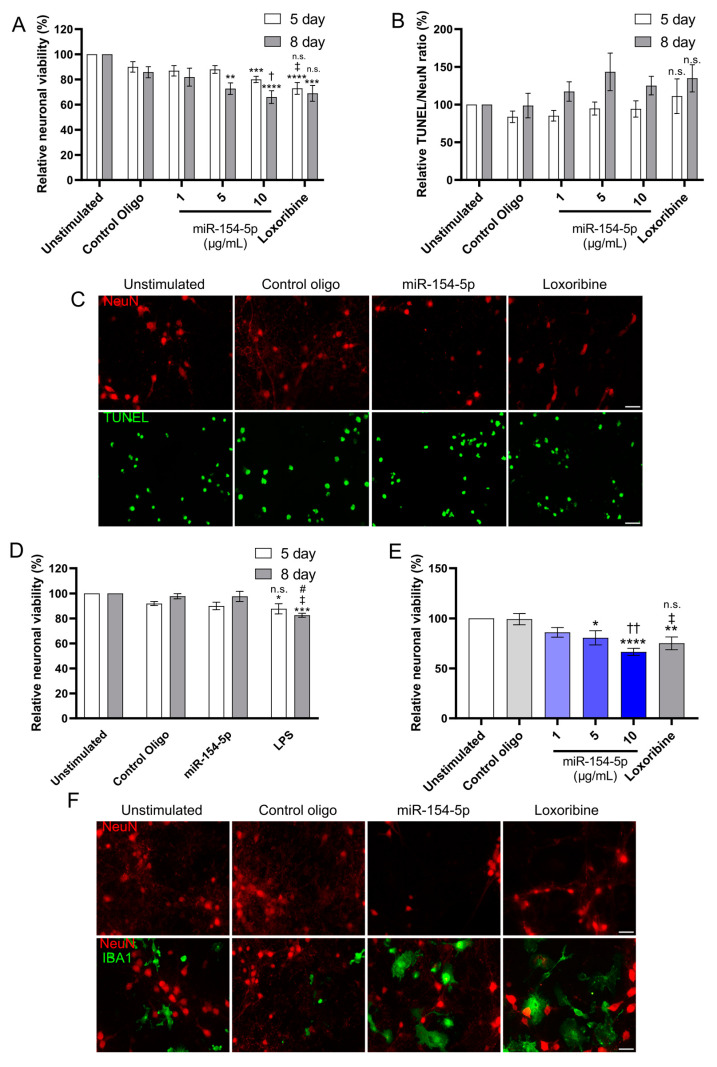
miR-154-5p causes dose- and time-dependent neuronal injury. (**A**,**B**) Primary neuron cultures were stimulated for 5 or 8 days with control oligo (10 μg/mL) as a negative control, indicated dose of miR-154-5p, or Loxoribine (1 mM) as a positive control, and cell toxicity was assessed by quantification of NeuN- or TUNEL-positivity. (**C**) Representative images based on percentage change to unstimulated for 8 day stimulation with control oligo (10 μg/mL), miR-154-5p (10 μg/mL) or Loxoribine (1 mM). (**D**) Primary *Tlr7^−/−^* neurons were stimulated for 5 or 8 days with control oligo (10 μg/mL) as a negative control, miR-154-5p (10 μg/mL), or LPS (100 ng/mL) as a positive control for microglial activation and immunostained with NeuN antibody for quantification. (**E**) Primary neuronal and microglial co-cultures were stimulated with indicated dose of miR-154-5p for 5 days and subsequently immunostained with NeuN antibody and quantified. (**F**) Representative images of co-cultures immunostained with NeuN and IBA1 antibodies after 5 days stimulation with miR-154-5p (10 μg/mL) or control oligo (10 μg/mL) and Loxoribine (1 mM) serving as controls. Scale bar represents 25 μm. Significance was determined using one-way ANOVA followed by Dunnett’s test compared to unstimulated control (*) or Control oligo (†) or by Student’s *t*-test comparing the highest dose miR-154-5p to positive control (n.s/#). *^/†/#^ *p* < 0.05, **^/‡^ *p* < 0.01, ***^/††^
*p* < 0.001, **** *p* < 0.0001.

**Figure 4 cells-13-00407-f004:**
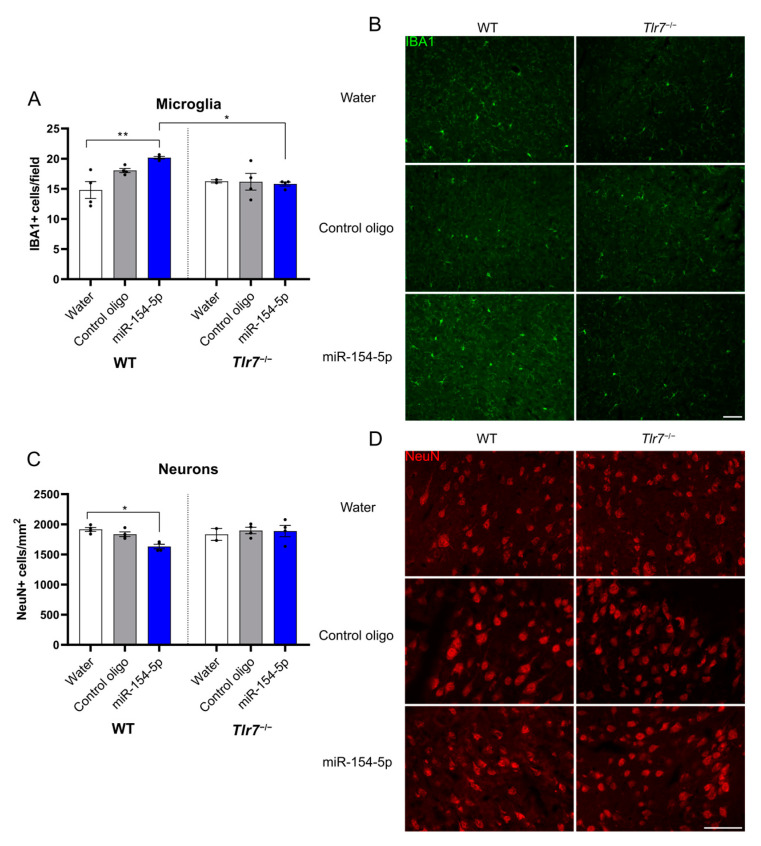
Intrathecal injection of miR-154-5p into mice induces microglial accumulation and neuronal injury. (**A**) Mice were intrathecally injected with miR-154-5p (10 μg/mL) or control oligo (10 μg/mL) and sacrificed after 3 days. Sections were immunostained with IBA1 and NeuN antibodies to mark microglia and neurons, respectively. Microglia were quantified in 6 total fields of both hemispheres at 20× magnification. (**B**) Representative images based on average number of microglia per field. Scale bar represents 50 μm. (**C**) Average number of NeuN-positive cells in 6 fields of both hemispheres. (**D**) Representative images based on average number of NeuN-positive cells per field. Scale bar represents 50 μm. Significance was determined by two-way ANOVA followed by Tukey’s multiple comparison test on all conditions to account for both treatment and genetic background. * *p* < 0.05, ** *p* < 0.01.

**Table 1 cells-13-00407-t001:** List of primers used in RT-qPCR.

Target Gene	Forward	Reverse
*β-Actin*	CCTGAACCCTAAGGCCAAC	GACAGCACAGCCTGGATGG
*MyD88*	ACCTGTGTCTGGTCCATTGCCA	GCTGAGTGCAAACTTGGTCTGG
*TICAM1*	ATCCATGCCAGGGCTGATGAAC	CGATGGCATCTTGGAGACAGTG
*IRF7*	CCTCTGCTTTCTAGTGATGCCG	CGTAAACACGGTCTTGCTCCTG
*p65*	TCCTGTTCGAGTCTCCATGCAG	GGTCTCATAGGTCCTTTTGCGC
*TNF*	GGTGCCTATGTCTCAGCCTCTT	GCCATAGAACTGATGAGAGGGAG
*IFNβ*	GCCTTTGCCATCCAAGAGATGC	ACACTGTCTGCTGGTGGAGTTC
*IL6*	TACCACTTCACAAGTCGGAGGC	CTGCAAGTGCATCATCGTTGTTC
*IRAK4*	CATACGCAACCTTAATGTGGGG	GGAACTGATTGTATCTGTCGTCG
*SARM1*	TTCCTTGGCTCCAGAAATGCT	GACCCTGAGTTCCTCCGGTA

## Data Availability

The datasets used and/or analyzed during the current study are available from the corresponding authors on reasonable request.

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
