# Peer review of "miR-154-5p Is a Novel Endogenous Ligand for TLR7 Inducing Microglial Activation and Neuronal Injury"

_cells, 2024, doi:10.3390/cells13050407_

Round 1
Reviewer 1 Report
Comments and Suggestions for Authors
1) In the introduction, the authors should discuss 2 types of glial activation - A1 and A2 types. Their features and markers.
2) Cell Lines. Authors must indicate from which passage the cells were used for experiments. How were cells examined for mycoplasma contamination?
3) Primary Cell Culture. Authors must clearly demonstrate that neuronal cell cultures were pure and did not contain astrocytes or microglia. For example, NeuN and GFAP staining. The same is true for microglial cell cultures. In addition, for mixed cell cultures, the ratio of microglia to neurons must be clearly stated. Simultaneous staining should be provided since the authors have these antibodies in their arsenal. This is the key question to this article.
4) Authors must justify the use of β-Actin as a reference gene. There is convincing evidence for the need to use two reference genes and for brain cells one of these genes should be OAZ
5) For all figures containing micrographs, it is necessary to provide images of cells in transmitted light (Bright-field microscopy), and the images must also be enlarged.
6) Figure 3 looks like a copy-paste page. Must be redone.
7) A separate conclusions section should be presented. Conclusions should be more meaningful than what is available at the end in the form of In conclusion...
Author Response
Reviewer #1
1) In the introduction, the authors should discuss 2 types of glial activation - A1 and A2 types. Their features and markers.
Response: We have now included a paragraph on the M1/M2 phenotype, as our study focuses on microglial activation, in the Introduction and Discussion sections where we discuss the microglial states and their association with both miRNAs in general and miR-154-5p (see pp. 2 (lines 76-82), 12 (lines 366-68)).
2) Cell Lines. Authors must indicate from which passage the cells were used for experiments. How were cells examined for mycoplasma contamination?
Response: Maximum passage number of cell line experiments (HEK cells, <20 passages) has now been indicated in the revised Methods section (section 2.2). Additionally, manufacturer’s recommendations on contamination prevention in these cells were performed and are now clearly indicated in the manuscript (see p. 3, lines 120-22).
3) Primary Cell Culture. Authors must clearly demonstrate that neuronal cell cultures were pure and did not contain astrocytes or microglia. For example, NeuN and GFAP staining. The same is true for microglial cell cultures. In addition, for mixed cell cultures, the ratio of microglia to neurons must be clearly stated. Simultaneous staining should be provided since the authors have these antibodies in their arsenal. This is the key question to this article.
Response: We generate enriched cortical neuronal cultures, microglial cultures, and co-cultures containing neurons and microglia, which are frequently tested for purity by immunostaining, on a regular basis in our laboratory. The purity of the enriched neuronal cultures used in our studies has been previously documented, demonstrating that these cell cultures do not contain significant numbers of glial cells [1]. In detail, based on immunocytochemical analysis with NeuN (marker for neurons), Iba-1 (microglial marker), and GFAP (astrocyte marker) antibodies and dependent on the developmental stage of the neurons, our enriched neuronal cultures contain 2.4% microglia, 1.0% astrocytes, and 0.2 oligodendrocytes (DIV 3 [1], as used in our current study). In addition to the representative images in Figure 3 of the original manuscript we have included images of microglial cultures immunostained with IBA-1 antibody and stained with DAPI below (Figure 1A). Also, enriched neuronal cultures used in our current study, immunostained with NeuN and Iba1 antibodies are shown (Figure 1B). While in Figure 1A all DAPI cells are positive for Iba-1 expression and therefore can be considered as microglia, only a minor number, if at all, of microglia (Iba-1+) and astrocytes (GFAP+) can be detected in the enriched neuronal cultures. These data are in line with our previous analyses
regarding the purity of the cell cultures, as described above.
Also, we have added the microglia:neuron ratio (1:8) in the Methods section (section 2.3, see p. 4, line 153) of the revised manuscript, as requested.
4) Authors must justify the use of β-Actin as a reference gene. There is convincing evidence for the need to use two reference genes and for brain cells one of these genes should be OAZ
Response: Indeed, there is evidence suggesting that β-actin may not be a suitable reference gene in all circumstances, particularly in human disease tissue or single animal tissue due to the individual variability of β-actin expression. However, in the experimental approach displayed in Figure 2 we primarily aim to assess the difference in signalling molecule expression between wild-type and Tlr7-/- microglia rather than to determine absolute expression values. Thus, we consider the usage of b-actin as a reference gene in our experimental approach as being sufficient and correct.
5) For all figures containing micrographs, it is necessary to provide images of cells in transmitted light (Bright-field microscopy), and the images must also be enlarged.
Response: As requested, we have included phase-contrast images of the cell cultures used in the respective experiments in the revised manuscript version (see new Supplemental Figure 1, Results section lines 287, 300). Also, we have enlarged the images throughout the manuscript.
6) Figure 3 looks like a copy-paste page. Must be redone.
Response: We thank the reviewer for bringing this mistake to our attention. It has now been corrected with a high-quality image (see revised Figure 3).
7) A separate conclusions section should be presented. Conclusions should be more meaningful than what is available at the end in the form of In conclusion...
Response: We have now included a separate Conclusion section in the revised manuscript, following the Discussion section, as requested (see p. 14, lines 457-67).
References
[1] Lehmann SM, Krüger C, Park B, Derkow K, Rosenberger K, Baumgart J, Trimbuch T, Eom G, Hinz M, Kaul D, Habbel P, Kälin R, Franzoni E, Rybak A, Nguyen D, Veh R, Ninnemann O, Peters O, Nitsch R, Heppner FL, Golenbock D, Schott E, Ploegh HL, Wulczyn FG, Lehnardt S (2012) An unconventional role for miRNA: let-7 activates Toll-like receptor 7 and causes neurodegeneration. Nat Neurosci 15, 827–835.
[2] Wallach T, Mossmann ZJ, Szczepek M, Wetzel M, Machado R, Raden M, Miladi M, Kleinau G, Krüger C, Dembny P, Adler D, Zhai Y, Kumbol V, Dzaye O, Schüler J, Futschik M, Backofen R, Scheerer P, Lehnardt S (2021) MicroRNA-100-5p and microRNA-298-5p released from apoptotic cortical neurons are endogenous Toll-like receptor 7/8 ligands that contribute to neurodegeneration. Mol Neurodegener 16, 80.
[3] Luo Z, Su R, Wang W, Liang Y, Zeng X, Shereen MA, Bashir N, Zhang Q, Zhao L, Wu K, Liu Y, Wu J (2019) EV71 infection induces neurodegeneration via activating TLR7 signaling and IL-6 production. PLOS Pathog 15, e1008142.
[4] Lehmann SM, Rosenberger K, Krüger C, Habbel P, Derkow K, Kaul D, Rybak A, Brandt C, Schott E, Wulczyn FG, Lehnardt S (2012) Extracellularly Delivered Single-Stranded Viral RNA Causes Neurodegeneration Dependent on TLR7. J Immunol 189, 1448–1458.
[5] Yu Liu H, Fen Hung Y, Ru Lin H, Li Yen T, Hsueh YP (2017) Tlr7 Deletion Selectively Ameliorates Spatial Learning but does not Influence beta Deposition and Inflammatory Response in an Alzheimers Disease Mouse Model. Neuropsychiatry 07, 509–521.
[6] Chen S, Wang X, Qian Z, Wang M, Zhang F, Zeng T, Li L, Gao L (2023) Exosomes from ADSCs ameliorate nerve damage in the hippocampus caused by post traumatic brain injury via the delivery of circ-Scmh1 promoting microglial M2 polarization. Injury 54, 110927.
[7] Sriram U, Xu J, Chain RW, Varghese L, Chakhtoura M, Bennett HL, Zoltick PW, Gallucci S (2014) IL-4 Suppresses the Responses to TLR7 and TLR9 Stimulation and Increases the Permissiveness to Retroviral Infection of Murine Conventional Dendritic Cells. PLoS ONE 9, e87668.
[8] Kim M, Jo H, Kwon Y, Jeong MS, Jung HS, Kim Y, Jeoung D (2021) MiR-154-5p-MCP1 Axis Regulates Allergic Inflammation by Mediating Cellular Interactions. Front Immunol 12,.

Reviewer 2 Report
Comments and Suggestions for Authors
It is an interesting study. The authors examined the effect of miR-154-5p as an endogenous ligand for TLR7 in inducing microglial activation and neuronal injury. They found that miR-154-5p acts as a direct activator of TLR7, leading to neuroinflammation and neuronal injury. However, microglia express several Toll-like receptors (TLRs; 1-9) that play crucial roles in sensing and responding to pathogens and a wide variety of endogenous molecules. In this study, the authors examined only two types of TLRs. Literature suggests that microglial activation is largely due to stimulation of TLR2, TLR4, and TLR9, characterized by the release of cytokines such as TNF-α, IL-1β, and IL-12 (Figure 1E also supports this). Whereas in this study, authors demonstrated that TLR7 also induces microglial activation and TNF-alpha levels. Hence, this study contributes to the advancement of our understanding of TLRs in microglial activation and neuroinflammation at least in the murine model. The authors would have measured other important proinflammatory cytokines such as IL-6, IFNs, and IL-1beta besides TNF-alpha.
Author Response
Reviewer #2
- It is an interesting study. The authors examined the effect of miR-154-5p as an endogenous ligand for TLR7 in inducing microglial activation and neuronal injury. They found that miR-154-5p acts as a direct activator of TLR7, leading to neuroinflammation and neuronal injury. However, microglia express several Toll-like receptors (TLRs; 1-9) that play crucial roles in sensing and responding to pathogens and a wide variety of endogenous molecules. In this study, the authors examined only two types of TLRs. Literature suggests that microglial activation is largely due to stimulation of TLR2, TLR4, and TLR9, characterized by the release of cytokines such as TNF-α, IL-1β, and IL-12 (Figure 1E also supports this). Whereas in this study, authors demonstrated that TLR7 also induces microglial activation and TNF-alpha levels. Hence, this study contributes to the advancement of our understanding of TLRs in microglial activation and neuroinflammation at least in the murine model. The authors would have measured other important proinflammatory cytokines such as IL-6, IFNs, and IL-1beta besides TNF-alpha.
Response: We would like to thank the reviewer for her/his comments. It is certainly well established that the stimulation of e.g. TLR2 and TLR4 causes the release of cytokines, such as IL-1β or TNF from microglia. In our current study the focus lies on miRNAs acting as signalling molecules for immune receptors. The TLR recognizing these small RNAs is indeed TLR7. Our data show that in consequence of this interaction, microglia express/release chemokines/cytokines. We demonstrate a direct interaction between the miRNA miR-154-5p and TLR7 in Figure 1E and Figure 2 showing a lack of response from Tlr7-/- cells exposed to the (extracellularly delivered) miRNA. Although we determined some additional cytokines expressed at the mRNA level (Figure 2), we agree that an in-depth characterisation of cytokine/chemokine release from microglia responding to miRNAs acting as signalling molecules for TLR7 would be valuable to better understand the miR-154-5p-induced TLR7 response, i) especially in a human system, ii) to compare human and murine responses, as well as iii) to compare TLR7 vs. TLR8 (another TLR sensing small ssRNA and miRNA) responses. We have included these thoughts in the revised Discussion section (see pp. 12-13).
References
[1] Lehmann SM, Krüger C, Park B, Derkow K, Rosenberger K, Baumgart J, Trimbuch T, Eom G, Hinz M, Kaul D, Habbel P, Kälin R, Franzoni E, Rybak A, Nguyen D, Veh R, Ninnemann O, Peters O, Nitsch R, Heppner FL, Golenbock D, Schott E, Ploegh HL, Wulczyn FG, Lehnardt S (2012) An unconventional role for miRNA: let-7 activates Toll-like receptor 7 and causes neurodegeneration. Nat Neurosci 15, 827–835.
[2] Wallach T, Mossmann ZJ, Szczepek M, Wetzel M, Machado R, Raden M, Miladi M, Kleinau G, Krüger C, Dembny P, Adler D, Zhai Y, Kumbol V, Dzaye O, Schüler J, Futschik M, Backofen R, Scheerer P, Lehnardt S (2021) MicroRNA-100-5p and microRNA-298-5p released from apoptotic cortical neurons are endogenous Toll-like receptor 7/8 ligands that contribute to neurodegeneration. Mol Neurodegener 16, 80.
[3] Luo Z, Su R, Wang W, Liang Y, Zeng X, Shereen MA, Bashir N, Zhang Q, Zhao L, Wu K, Liu Y, Wu J (2019) EV71 infection induces neurodegeneration via activating TLR7 signaling and IL-6 production. PLOS Pathog 15, e1008142.
[4] Lehmann SM, Rosenberger K, Krüger C, Habbel P, Derkow K, Kaul D, Rybak A, Brandt C, Schott E, Wulczyn FG, Lehnardt S (2012) Extracellularly Delivered Single-Stranded Viral RNA Causes Neurodegeneration Dependent on TLR7. J Immunol 189, 1448–1458.
[5] Yu Liu H, Fen Hung Y, Ru Lin H, Li Yen T, Hsueh YP (2017) Tlr7 Deletion Selectively Ameliorates Spatial Learning but does not Influence beta Deposition and Inflammatory Response in an Alzheimers Disease Mouse Model. Neuropsychiatry 07, 509–521.
[6] Chen S, Wang X, Qian Z, Wang M, Zhang F, Zeng T, Li L, Gao L (2023) Exosomes from ADSCs ameliorate nerve damage in the hippocampus caused by post traumatic brain injury via the delivery of circ-Scmh1 promoting microglial M2 polarization. Injury 54, 110927.
[7] Sriram U, Xu J, Chain RW, Varghese L, Chakhtoura M, Bennett HL, Zoltick PW, Gallucci S (2014) IL-4 Suppresses the Responses to TLR7 and TLR9 Stimulation and Increases the Permissiveness to Retroviral Infection of Murine Conventional Dendritic Cells. PLoS ONE 9, e87668.
[8] Kim M, Jo H, Kwon Y, Jeong MS, Jung HS, Kim Y, Jeoung D (2021) MiR-154-5p-MCP1 Axis Regulates Allergic Inflammation by Mediating Cellular Interactions. Front Immunol 12,.
Reviewer 3 Report
Comments and Suggestions for Authors
This research article, "miR-154-5p is a novel endogenous ligand for TLR7 inducing microglial activation and neuronal injury," unveils miR-154-5p's role in neuroinflammation and neurodegeneration. Traditionally, microRNAs (miRNAs) are known for regulating gene expression, but this study reveals their capability to act as ligands for Toll-like receptors (TLRs), expanding their functional repertoire. Through a combination of in vitro and in vivo experiments, the authors demonstrate that miR-154-5p activates TLR7, leading to microglial activation and neuronal damage. This discovery highlights a new mechanism of neuroinflammation and suggests miR-154-5p as a potential therapeutic target in neurodegenerative diseases.
The findings show that miR-154-5p's interaction with TLR7 triggers inflammatory responses in microglia, causing the release of pro-inflammatory cytokines and resulting in neuronal injury. This direct link between miR-154-5p, microglial activation, and neurodegeneration underscores its significance in cell-autonomous neurotoxicity. The study not only contributes to understanding the complex interplay between miRNAs and immune signaling in the central nervous system but also opens new paths for investigating miRNA roles in neuroinflammation and neurodegeneration, offering insights into novel therapeutic approaches for these conditions.
To enhance the quality of your manuscript, the following improvements are suggested:
Introduction
1. The introduction provides a good overview of the role of the innate immune system, specifically TLRs, in neurodegenerative diseases and introduces miRNAs' role in this context effectively. However, it could benefit from a more structured breakdown, perhaps by introducing headings or bullet points to distinguish between the background on TLRs, the role of miRNAs, and the specific focus on miR-154-5p. This could enhance readability and make the narrative flow more logical.
2. While the introduction mentions the role of TLR7 in detecting ssRNA and miRNAs acting as ligands for TLR7, it would be beneficial to include more details about the underlying mechanisms. For instance, how does miR-154-5p's interaction with TLR7 contribute to neuroinflammation and neurodegeneration? Are there specific signaling pathways activated? More specificity here could strengthen the rationale for focusing on miR-154-5p.
3. The manuscript claims to identify miR-154-5p as an endogenous ligand for TLR7, which is an important finding. However, the introduction should better contextualize this discovery within the existing literature. How does this add to or challenge current understanding? Are there conflicting reports on miR-154-5p's role in neurological diseases?
4. The introduction does a good job referencing key studies, but it should ensure that the most current and relevant literature is cited. For instance, are there any very recent studies (post the latest cited references) that have furthered understanding of miR-154-5p's role or TLR7 signaling in neurodegenerative diseases?
5. While the introduction outlines the background well, it could more explicitly state the study's hypothesis or research question. What specific gaps in the current understanding does this study aim to address? Making this clearer could help frame the importance of the research and its potential impact.
6. Ensure that the interpretation of previous findings and the presentation of the new data are balanced. The introduction seems to heavily imply causality between miR-154-5p dysregulation and neurodegenerative diseases. While suggestive, it's crucial to acknowledge the complexity of these diseases and the multifactorial elements involved, potentially tempering conclusions until further supported by the study's findings.
7. The manuscript uses technical terminology appropriately but could benefit from briefly defining or explaining key concepts for a broader readership. For example, a brief explanation of what constitutes a "non-canonical function" for miRNAs could make the introduction more accessible to non-specialists.
Results
8. The results are comprehensive and detailed, which is commendable. However, the presentation could be made more digestible by breaking down complex findings into smaller, more focused paragraphs. This could help readers understand the significance of each finding more clearly.
9. It's crucial to mention the statistical tests used to determine the significance of the observed differences, including details on the p-values, statistical significance thresholds, and any corrections for multiple comparisons. This will provide clarity on how conclusions were drawn from the data.
10. While the experimental approaches are described, adding more details about the methodology (e.g., concentrations of miR-154-5p used, duration of incubation times, and specifics of the cell culture conditions) would improve reproducibility and clarity.
11. The manuscript could benefit from a comparison between the effects of miR-154-5p and other known TLR7/8 ligands. This would contextualize the relative potency and specificity of miR-154-5p in activating these receptors.
12. While the results show that miR-154-5p activates TLR7/8 and induces signaling changes and neuronal injury, further discussion on the mechanistic insights into how miR-154-5p leads to these downstream effects would be valuable. Are there specific signaling pathways or molecular interactions that are particularly affected?
13. The unchanged levels of TRIF and MYD88 adapter proteins in stimulated vs. unstimulated cells warrant a deeper discussion. The implications of these findings on the traditional understanding of TLR signaling pathways could be an interesting point of exploration.
14. While the results section is robust, suggesting directions for future research based on these findings could be helpful. For instance, exploring therapeutic interventions that could modulate miR-154-5p levels or block its interaction with TLR7/8 might be a promising area.
15. Mention any steps taken to ensure the specificity of miR-154-5p's effects, such as controls for potential off-target effects of miRNA mimics or inhibitors used in the experiments.
Discussion
16. You've done an excellent job of situating miR-154-5p within the broader landscape of miRNA research and its implications for neurodegenerative diseases. Emphasizing the novelty of your findings - specifically miR-154-5p's direct activation of TLR7 and its dual role in microglial activation and neuronal injury - early in the discussion would help underscore the significance of your work.
17. Your discussion makes a strong case for the therapeutic potential of targeting miR-154-5p in diseases characterized by neuroinflammation and neurodegeneration. Expanding on the practical aspects of such interventions, including potential strategies for modulating miR-154-5p levels or function, could be very enlightening. This might include antisense oligonucleotides, small molecule inhibitors, or other miRNA-based therapies.
18. While your discussion provides a comprehensive overview of your findings, addressing potential limitations of your study could strengthen it. For instance, discussing any constraints in the extrapolation of in vitro findings to in vivo contexts, or the challenges in targeting miRNA pathways without affecting their physiological roles, would provide a balanced view.
19. Your discussion on the possible mechanisms of TLR7-mediated toxicity in neurons and the role of microglia in neurodegeneration is intriguing. Speculating on how miR-154-5p might interact with other cellular pathways or contribute to the pathology of neurodegenerative diseases could spur future research directions. However, it's crucial to frame these speculations carefully, noting the preliminary nature of such hypotheses.
20. Placing miR-154-5p's role in the context of other miRNAs known to interact with TLR7 and TLR8, and discussing the specificity, potency, or therapeutic implications of targeting miR-154-5p versus these other miRNAs, would provide valuable perspective on the potential "druggability" of miR-154-5p.
21. You've outlined several intriguing future research questions. Expanding on these by suggesting specific experimental approaches or models to address these questions could be very helpful for guiding the next steps in this research area. This might include the use of transgenic animal models, longitudinal studies to assess the kinetics of miR-154-5p in neurodegeneration, or high-throughput screening for modulators of miR-154-5p expression.
22. Given the complex nature of neurodegenerative diseases and the multifaceted roles of miRNAs, encouraging interdisciplinary collaboration between neuroscientists, immunologists, molecular biologists, and bioinformaticians could foster the development of innovative therapeutic strategies.
Comments on the Quality of English Language1. Ensure consistent use of specific terms and phrases related to neurogenesis and regeneration throughout the manuscript. This includes the standardized use of abbreviations after they are first introduced.
2. Review the manuscript for any grammatical errors, including subject-verb agreement, proper use of articles ("a", "an", "the"), and correct tense usage. Although the scientific content appears strong, minor grammatical errors can detract from the manuscript's overall impact.
Author Response
Reviewer #3
Introduction
- The introduction provides a good overview of the role of the innate immune system, specifically TLRs, in neurodegenerative diseases and introduces miRNAs' role in this context effectively. However, it could benefit from a more structured breakdown, perhaps by introducing headings or bullet points to distinguish between the background on TLRs, the role of miRNAs, and the specific focus on miR-154-5p. This could enhance readability and make the narrative flow more logical.
Response: We have now included additional paragraphs in the Introduction to delineate the sections more clearly, as suggested; background on TLRs, general miRNA background, mir-154 background, and finally a brief summary of the results (see pp. 1-3).
- While the introduction mentions the role of TLR7 in detecting ssRNA and miRNAs acting as ligands for TLR7, it would be beneficial to include more details about the underlying mechanisms. For instance, how does miR-154-5p's interaction with TLR7 contribute to neuroinflammation and neurodegeneration? Are there specific signaling pathways activated? More specificity here could strengthen the rationale for focusing on miR-154-5p.
Response: We have included some additional information in the Introduction regarding the binding / interaction between miRNAs and TLR7, particularly referencing the underlying sequence specificity of the miRNAs acting as a signaling molecules (see p. 2, lines 68-73). Our group has previously shown that direct TLR7 activation in neurons by extracellular let-7, miR-298, and miR-100, abundantly expressed in the brain, leads to activation of neuronal caspase-3 and subsequent neurodegeneration. Also, neuronal TLR7 activation by let-7 involves MyD88 and IRAK 4 signalling [1,2]. However, the detailed, further signalling cascade downstream of TLR7 remains unresolved. TLR7 activation in microglia by extracellular let-7, miR-100, and miR-298 results in canonical TLR signalling including NF-kB activation, and ultimately in the release of potentially neurotoxic chemokines and cytokines [1,2]. Although analysing the expression of some of the canonical TLR7 signalling members in microglia in response to miR-154-5p, further studies are required to understand how miRNAs acting as ligands affect TLR7 signalling. We have included these thoughts in the revised Discussion section (see p. 12, lines 339-48).
- The manuscript claims to identify miR-154-5p as an endogenous ligand for TLR7, which is an important finding. However, the introduction should better contextualize this discovery within the existing literature. How does this add to or challenge current understanding? Are there conflicting reports on miR-154-5p's role in neurological diseases?
Response: We have included some additional studies on the role of miR-154-5p that are relevant to the current study and help to contextualize our findings, as suggested. In particular, we have cited two studies exploring the role of miR-154-5p in vascular dementia and prevention of cell dysfunction and miR-154-5p preventing a shift towards anti-inflammatory microglial states (see p. 2, lines 88-93).
- The introduction does a good job referencing key studies, but it should ensure that the most current and relevant literature is cited. For instance, are there any very recent studies (post the latest cited references) that have furthered understanding of miR-154-5p's role or TLR7 signaling in neurodegenerative diseases?
Response: As outlined in the original manuscript and above, the mechanistic role of miR-154-5p is poorly understood, particularly in neurodegenerative disease. However, it has been shown that miR-154-5p is involved in the reduction of neuronal apoptosis in a cellular model of epilepsy but is also partially responsible for cellular dysfunction in vascular dementia. Thus, the role of miR-154-5p may be highly dependent on the animal/disease model and context it is being applied. This is similar for TLR7, which has been shown to contribute to neuronal cell death in a cell-autonomous fashion and via increased inflammation, e.g. in the context of viral infections [3,4]. However, in an Alzheimer’s disease (AD) model, deletion of TLR7 had no effect on inflammation, but did rescue some cognitive deficits [5]. These data may indicate a disparity between acute and chronic TLR7 function. Our study primarily focuses on short-term effects induced by TLR7 activation through extracellular miRNA. Further research on longer term effects of miR-154-5p-induced TLR7 activation would be needed to determine its effects in detail and analyse the dependency on a specific (disease) context. As mentioned above, this data and thoughts have now been included in the revised Introduction (p. 2, lines 47-57, 88-93).
- 5. While the introduction outlines the background well, it could more explicitly state the study's hypothesis or research question. What specific gaps in the current understanding does this study aim to address? Making this clearer could help frame the importance of the research and its potential impact.
Response: As mentioned in the previous section, we now have included additional information on miR-154 and the role of TLR7 in neurodegeneration to add more context to the study. We have also more explicitly stated the research aims of the paper (see p. 2, lines 93-96), as suggested.
- Ensure that the interpretation of previous findings and the presentation of the new data are balanced. The introduction seems to heavily imply causality between miR-154-5p dysregulation and neurodegenerative diseases. While suggestive, it's crucial to acknowledge the complexity of these diseases and the multifactorial elements involved, potentially tempering conclusions until further supported by the study's findings.
Response: We agree with the reviewer that the respective disease contexts and the multifactorial elements involved are likely highly complex. Also, we fully agree that our data constitute no basis for causality between miR-154-5p-induced TLR7 signalling and (any) neurodegenerative disease. For clarification, we have now made changes to frame the effects we see less strongly in terms of such a potential causality and to provide a more balanced view of the results (see revised Discussion, pp. 12-14).
- The manuscript uses technical terminology appropriately but could benefit from briefly defining or explaining key concepts for a broader readership. For example, a brief explanation of what constitutes a "non-canonical function" for miRNAs could make the introduction more accessible to non-specialists.
Response: As suggested, we have reemphasised the definition ‘non-canonical’ terminology in the revised manuscript (p. 2, 64-66).
Results
- The results are comprehensive and detailed, which is commendable. However, the presentation could be made more digestible by breaking down complex findings into smaller, more focused paragraphs. This could help readers understand the significance of each finding more clearly.
Response: We thank the reviewer for her/his comment. We have now gone through the larger Results sections (pp. 6-11) and included additional paragraphs to improve readability of the revised manuscript.
- It's crucial to mention the statistical tests used to determine the significance of the observed differences, including details on the p-values, statistical significance thresholds, and any corrections for multiple comparisons. This will provide clarity on how conclusions were drawn from the data.
Response: We have ensured that all statistical tests are stated in the relevant Figure legends as well as in the statistics section of the Methods section (see Methods, 2.1.2, p. 5, lines 225-231). In brief: all HEK reporter assays and ELISA comparing WT and Tlr7-/- were tested using a t-test comparing the TLR7/8 expressing cells and their parental control line. All other in vitro assays were assessed using a one-way ANOVA followed by Dunnett’s test for multiple comparisons. The in vivo experiment was tested using a two-way ANOVA followed by Tukey’s test for multiple comparison.
- 10. While the experimental approaches are described, adding more details about the methodology (e.g., concentrations of miR-154-5p used, duration of incubation times, and specifics of the cell culture conditions) would improve reproducibility and clarity.
Response: We have double-checked that all concentrations and experimental durations are indicated either in the Figure directly or in the Figure legend.
- The manuscript could benefit from a comparison between the effects of miR-154-5p and other known TLR7/8 ligands. This would contextualize the relative potency and specificity of miR-154-5p in activating these receptors.
Response: All in vitro studies were performed using the TLR7 agonist loxoribine and/or TLR7/8 agonist R848 as a positive control for cell activation (excluding Tlr7-/- cell activation, where LPS was used) to compare the effects of the condition miR-154-5p with an established TLR7/8 agonist. As suggested, in the revised manuscript version we have compared the condition miR-154-5p (if dose responses were investigated, the highest dose was used for comparison) with the respective TLR agonist and indicated the result from the statistical significance testing in the respective Figure (see revised Figures 1 and 3).
- While the results show that miR-154-5p activates TLR7/8 and induces signaling changes and neuronal injury, further discussion on the mechanistic insights into how miR-154-5p leads to these downstream effects would be valuable. Are there specific signaling pathways or molecular interactions that are particularly affected?
Response: As outlined above, very little is known on the specifics of miR-154-5p-induced signalling cascade. There is some data describing the ability of miR-154-5p to reverse a circ-Scmh1-induced microglial polarisation from M1 to M2 via STAT6 binding[6], which may alter TLR7 signalling [7]. Also, miR-154-5p seems to be involved in allergic inflammation mediated by MCP-1 [8]. However, beyond this, particularly if we consider miR-154-5p acting as a TLR ligand, rather than modulating signalling pathways via gene suppression/regulation at the post-transcriptional level, the detailed mechanisms are essentially a novel field of study for future research. We have included these thoughts in the revised Discussion section (pp. 12-13, lines 363-82).
- 13. The unchanged levels of TRIF and MYD88 adapter proteins in stimulated vs. unstimulated cells warrant a deeper discussion. The implications of these findings on the traditional understanding of TLR signaling pathways could be an interesting point of exploration.
Response: The alteration of some signalling components but not others such as MYD88 and TRIF is certainly of interest. As outlined above, at this point there is little research on the effect of miRNAs acting as ligands on the expression of TLR signalling components and what the consequences of such changes are. We plan on further investigating the signalling effects of miRNAs in the future by utilising e.g. MYD88 knockout mice and viral overexpression of miRNAs to see if a permanent increase in miRNA production will induce signalling changes or compensatory mechanisms. However, currently that is beyond the scope of this paper. We have included a statement on this in the revised Discussion section (pp. 12-13, lines 378-382).
- 14. While the results section is robust, suggesting directions for future research based on these findings could be helpful. For instance, exploring therapeutic interventions that could modulate miR-154-5p levels or block its interaction with TLR7/8 might be a promising area.
Response: We agree that further research on the specific modulation of miR-154-5p in the CNS is certainly of great interest and necessary, particularly in the context of neurodegenerative disease. Also, addressing the question how chronic inflammation could alter the effects of miR-154-5p up/downregulation in the CNS would be vital information. In our current study we focused on the effects induced by extracellularly delivered miR-154-5p but assessing the effect of miR-154-5p reduction in the CNS using an LNA-based miRNA inhibitor to confirm if we can reverse or prevent the effects, is the next step. We have included a paragraph on this in the revised Discussion section (p. 13, lines 414-17).
- 15. Mention any steps taken to ensure the specificity of miR-154-5p's effects, such as controls for potential off-target effects of miRNA mimics or inhibitors used in the experiments.
Response: The primary method used to control for off-target effects was the control oligonucleotide included in all experiments. As outlined above, this oligonucleotide contains a mutated sequence derived from the miRNA let-7b (containing six nucleotide exchanges in the central and 3′ regions, outside of the 5′ let-7 seed sequence that is important for post-transcriptional silencing) previously established in our lab, which does not activate TLR7 signalling and with no known human or mouse targets. This control helps to determine if there is a sequence-specific effect of miR-154-5p, rather than simply a response to extracellular RNA in the system [1]. We have included this information in the revised manuscript (see p. 6, lines 243-46). In our current study we did not use miR-154-5p inhibitors but as aforementioned this is the next logical step and planned in our laboratory.
Discussion
- 16. You've done an excellent job of situating miR-154-5p within the broader landscape of miRNA research and its implications for neurodegenerative diseases. Emphasizing the novelty of your findings - specifically miR-154-5p's direct activation of TLR7 and its dual role in microglial activation and neuronal injury - early in the discussion would help underscore the significance of your work.
Response: As suggested, we have now clearly outlined our findings at the beginning of the revised Discussion section (see p. 12, lines 326-27).
- Your discussion makes a strong case for the therapeutic potential of targeting miR-154-5p in diseases characterized by neuroinflammation and neurodegeneration. Expanding on the practical aspects of such interventions, including potential strategies for modulating miR-154-5p levels or function, could be very enlightening. This might include antisense oligonucleotides, small molecule inhibitors, or other miRNA-based therapies.
Response: As suggested, we have included some additional commentary on the more practical and clinical aspects of targeting miR-154-5p or miRNAs in general (see revised Discussion section, p. 14, lines 441-47).
- While your discussion provides a comprehensive overview of your findings, addressing potential limitations of your study could strengthen it. For instance, discussing any constraints in the extrapolation of in vitro findings to in vivo contexts, or the challenges in targeting miRNA pathways without affecting their physiological roles, would provide a balanced view.
Response: This is a valuable point as currently the idea of using miRNAs as a target to intervention is very much in its infancy. Since the production of miRNAs is usually a highly controlled situation, one certainly must be careful in altering it. Changes in gene expression through removal/addition of a given miRNA could have dramatic long-term and undesired effects. We have included these thoughts in the revised Discussion (p. 13, lines 387-91).
- 19. Your discussion on the possible mechanisms of TLR7-mediated toxicity in neurons and the role of microglia in neurodegeneration is intriguing. Speculating on how miR-154-5p might interact with other cellular pathways or contribute to the pathology of neurodegenerative diseases could spur future research directions. However, it's crucial to frame these speculations carefully, noting the preliminary nature of such hypotheses.
Response: As suggested, we have included some data regarding other aspects of miR-154-5p’s role and other pathway interactions (as discussed above, see Discussion section, p. 12, lines 354-62). In addition, we have carefully framed speculations, whenever necessary, in the revised manuscript.
- 20. Placing miR-154-5p's role in the context of other miRNAs known to interact with TLR7 and TLR8, and discussing the specificity, potency, or therapeutic implications of targeting miR-154-5p versus these other miRNAs, would provide valuable perspective on the potential "druggability" of miR-154-5p.
Response: Currently, we do not have data on regarding the specificity of miR-154-5p’s role over other TLR7-binding miRNAs. Our lab has found similar effects as discussed in this paper from other miRNAs, including let-7b [1], miR-100, and miR-298 [2]. As of now we can only confirm that TLR7 is required for miRNA-induced effects observed, and we do not know whether the specific miRNA causes identical TLR7-based responses. We have included this point in the revised Discussion section (p. 14, lines 441-47) to frame the potential of miR-154-5p’s ‘druggability’, as suggested.
- 21. You've outlined several intriguing future research questions. Expanding on these by suggesting specific experimental approaches or models to address these questions could be very helpful for guiding the next steps in this research area. This might include the use of transgenic animal models, longitudinal studies to assess the kinetics of miR-154-5p in neurodegeneration, or high-throughput screening for modulators of miR-154-5p expression.
Response: As suggested by the reviewer, we now have included a paragraph on the potential uses of transgenic animal models, longitudinal studies to assess the miR-154-5p kinetics in neurodegeneration, or high-throughput screening for modulators of miR-154-5p expression in the revised Discussion section (p. 13, lines 414-22).
- 22. Given the complex nature of neurodegenerative diseases and the multifaceted roles of miRNAs, encouraging interdisciplinary collaboration between neuroscientists, immunologists, molecular biologists, and bioinformaticians could foster the development of innovative therapeutic strategies.
Response: We fully agree with the reviewer’s statement and have included this thought in the Conclusion of the revised paper (see p. 14, lines 462-67).
Comments on the Quality of English Language
1.Ensure consistent use of specific terms and phrases related to neurogenesis and regeneration throughout the manuscript. This includes the standardized use of abbreviations after they are first introduced.
Response: We have thoroughly revised the manuscript text, as requested.
2.Review the manuscript for any grammatical errors, including subject-verb agreement, proper use of articles ("a", "an", "the"), and correct tense usage. Although the scientific content appears strong, minor grammatical errors can detract from the manuscript's overall impact.
Response: We have carefully revised the manuscript, particularly focussing on the consistent use of technical terms and grammar.
References
[1] Lehmann SM, Krüger C, Park B, Derkow K, Rosenberger K, Baumgart J, Trimbuch T, Eom G, Hinz M, Kaul D, Habbel P, Kälin R, Franzoni E, Rybak A, Nguyen D, Veh R, Ninnemann O, Peters O, Nitsch R, Heppner FL, Golenbock D, Schott E, Ploegh HL, Wulczyn FG, Lehnardt S (2012) An unconventional role for miRNA: let-7 activates Toll-like receptor 7 and causes neurodegeneration. Nat Neurosci 15, 827–835.
[2] Wallach T, Mossmann ZJ, Szczepek M, Wetzel M, Machado R, Raden M, Miladi M, Kleinau G, Krüger C, Dembny P, Adler D, Zhai Y, Kumbol V, Dzaye O, Schüler J, Futschik M, Backofen R, Scheerer P, Lehnardt S (2021) MicroRNA-100-5p and microRNA-298-5p released from apoptotic cortical neurons are endogenous Toll-like receptor 7/8 ligands that contribute to neurodegeneration. Mol Neurodegener 16, 80.
[3] Luo Z, Su R, Wang W, Liang Y, Zeng X, Shereen MA, Bashir N, Zhang Q, Zhao L, Wu K, Liu Y, Wu J (2019) EV71 infection induces neurodegeneration via activating TLR7 signaling and IL-6 production. PLOS Pathog 15, e1008142.
[4] Lehmann SM, Rosenberger K, Krüger C, Habbel P, Derkow K, Kaul D, Rybak A, Brandt C, Schott E, Wulczyn FG, Lehnardt S (2012) Extracellularly Delivered Single-Stranded Viral RNA Causes Neurodegeneration Dependent on TLR7. J Immunol 189, 1448–1458.
[5] Yu Liu H, Fen Hung Y, Ru Lin H, Li Yen T, Hsueh YP (2017) Tlr7 Deletion Selectively Ameliorates Spatial Learning but does not Influence beta Deposition and Inflammatory Response in an Alzheimers Disease Mouse Model. Neuropsychiatry 07, 509–521.
[6] Chen S, Wang X, Qian Z, Wang M, Zhang F, Zeng T, Li L, Gao L (2023) Exosomes from ADSCs ameliorate nerve damage in the hippocampus caused by post traumatic brain injury via the delivery of circ-Scmh1 promoting microglial M2 polarization. Injury 54, 110927.
[7] Sriram U, Xu J, Chain RW, Varghese L, Chakhtoura M, Bennett HL, Zoltick PW, Gallucci S (2014) IL-4 Suppresses the Responses to TLR7 and TLR9 Stimulation and Increases the Permissiveness to Retroviral Infection of Murine Conventional Dendritic Cells. PLoS ONE 9, e87668.
[8] Kim M, Jo H, Kwon Y, Jeong MS, Jung HS, Kim Y, Jeoung D (2021) MiR-154-5p-MCP1 Axis Regulates Allergic Inflammation by Mediating Cellular Interactions. Front Immunol 12,.
Round 2
Reviewer 1 Report
Comments and Suggestions for Authors
The authors have done a great job. Almost all my comments have been taken into account. The question of using several reference genes has not been resolved; the authors could easily redo the analysis using new primers. However, this fact is not critical
Reviewer 3 Report
Comments and Suggestions for Authors
I would like to extend my heartfelt thanks to the authors for their thorough and meticulous revisions. The manuscript has significantly improved since its first submission, demonstrating the authors' earnest dedication to addressing the issues previously identified.
Comments on the Quality of English LanguageMinor editing of English language required